# Long Non-Coding RNAs in the Control of Gametogenesis: Lessons from Fission Yeast

**DOI:** 10.3390/ncrna7020034

**Published:** 2021-06-11

**Authors:** Vedrana Andric, Mathieu Rougemaille

**Affiliations:** 1Institute for Integrative Biology of the Cell (I2BC), Université Paris-Saclay, CEA, CNRS, 91198 Gif-sur-Yvette, France; 2Institute Curie, PSL Research University, CNRS UMR3215, INSERM U934, 75005 Paris, France; vedrana.andric@curie.fr

**Keywords:** fission yeast, gametogenesis, gene expression, long non-coding RNAs, mitosis to meiosis transition

## Abstract

Long non-coding RNAs (lncRNAs) contribute to cell fate decisions by modulating genome expression and stability. In the fission yeast *Schizosaccharomyces pombe*, the transition from mitosis to meiosis results in a marked remodeling of gene expression profiles, which ultimately ensures gamete production and inheritance of genetic information to the offspring. This key developmental process involves a set of dedicated lncRNAs that shape cell cycle-dependent transcriptomes through a variety of mechanisms, including epigenetic modifications and the modulation of transcription, post-transcriptional and post-translational regulations, and that contribute to meiosis-specific chromosomal events. In this review, we summarize the biology of these lncRNAs, from their identification to mechanism of action, and discuss their regulatory role in the control of gametogenesis.

## 1. Introduction

The largest share of eukaryotic genomes consists of elements with no coding potential. The development of high-throughput sequencing technologies has profoundly modified our vision on genome expression and unveiled the widespread nature of transcription, including in regions originally thought to be refractory to RNA synthesis (e.g., heterochromatin, intergenic regions). A myriad of non-coding RNAs (ncRNAs) is indeed generated in eukaryotic cells, encompassing heterogenous transcripts that can be broadly divided into two classes based on their length: small ncRNAs (e.g., miRNAs, siRNAs) that are below 200 nucleotides, as opposed to long ncRNAs (lncRNAs). Generally exhibiting poor sequence conservation and low expression levels, lncRNAs originate from intergenic regions, bidirectional transcription or overlap protein-coding genes in sense or antisense orientations [1,2]. They can act in *cis*, regulating the expression of adjacent genes, or in *trans*, exerting their role at distant genomic sites or in dedicated subcellular compartments [2,3]. Essentially involved in all steps in gene expression, lncRNAs can function as guides to recruit dedicated factors (e.g., chromatin-remodeling and modifying enzymes) to specific genomic loci, as sponges, titrating away RNA or proteins from their natural targets to modulate mRNA fate (e.g., splicing, translation, degradation), or as scaffolds to bring factors in spatial proximity (e.g., within nuclear bodies) [3,4,5,6]. Additional lncRNAs do not carry a function on their own, reflecting by-products of regulatory transcription, while others encode functionally relevant micropeptides [4,7,8]. LncRNAs are also known to partake in genome architecture and stability, favoring chromatin organization into defined territories and serving as signals and/or templates for DNA repair mechanisms, respectively [5,6,9,10,11]. Highlighting their biological relevance, lncRNAs are involved in key cellular processes, such as metabolism, differentiation and development, and they have been linked to various human diseases including cancer [6,11,12,13]. The majority also exhibit a tissue-specific expression, supporting a role in cell identity. Deciphering the role of lncRNAs in cell fate decisions is therefore essential to understand the molecular bases underlying developmental disorders and cancer.

Common to all sexually reproducing eukaryotes, meiosis is a highly conserved developmental process essential for genetic diversity within a population and the production of a normal offspring. The core steps of this specialized cell cycle consist of a single round of DNA replication followed by two sequential cellular divisions, which results in the generation of haploid cells, also named gametes, from diploid progenitors. In metazoans, meiosis follows the sexual maturity of an individual, while it is generally initiated in response to environmental cues to survive unfavorable conditions in unicellular eukaryotes. Among these is the fission yeast *Schizosaccharomyces pombe*, a powerful genetically tractable model organism for the study of the cellular pathways and molecular mechanisms underlying meiosis and sexual differentiation. When exposed to prolonged nutritional starvation, two *S. pombe* haploid cells of the opposite mating types, *h+* and *h−,* conjugate to form a zygote that subsequently enters the meiotic cell cycle [14]. Following pre-meiotic DNA synthesis, zygotes undergo meiotic divisions, finally giving rise to an ascus containing four haploid spores. The transition from mitosis to meiosis is accompanied by profound changes in gene expression profiles, whereby sequential transcriptional waves controlled by dedicated transcription factors orchestrate the correct timing of the meiotic program [15,16]. Gene expression during sexual differentiation is further tuned by chromatin modifications at specific meiotic genes, post-transcriptional mechanisms involved in the regulation of meiotic RNA stability and post-translational modifications of key meiotic effectors [14,17]. All these regulatory events are coordinated to ensure the irreversible commitment of cells to meiosis.

Early and more recent works have unraveled the requirement for lncRNAs in the control of gametogenesis in fission yeast. Remarkably, these lncRNAs display radically different modes of action to control meiotic gene expression and are themselves subjected to tight regulations. While certain functions are essential to prevent untimely meiosis during the mitotic cell cycle, others are critical for the induction and progression of sexual differentiation when appropriate conditions are met. Below, we summarize the current knowledge about the biogenesis of these lncRNAs, their respective targets and mechanism of action, as well as the contribution of protein cofactors to functional outputs.

## 2. The lncRNA *rse1* Guides Chromatin Modifiers to Repress in *cis* the Expression of the Meiosis-Specific Transcription Factor Ste11 in Mitotic Cells

*S. pombe* cells respond to environmental changes, including the lack of carbon or nitrogen sources, the presence of mating pheromones, as well as heat shock or oxidative stresses, by activating dedicated transduction cascades [14,18]. These signaling pathways ultimately converge to the activation of a master regulator of meiosis, the high mobility group (HMG) family transcription factor Ste11 [18,19,20]. The regulation of Ste11 abundance is essential for sexual differentiation as its deletion leads to complete sterility, while its over-expression triggers uncontrolled meiosis. Through the recognition of *cis*-acting elements within promoters (i.e., TR boxes) by its N-terminal HMG domain, Ste11 mediates the transcriptional induction of several genes involved in premeiotic DNA replication, mating and meiosis [19,21]. Ste11 also stimulates its own expression and targets key regulators of sexual differentiation, including the RNA-binding protein Mei2 and the meiosis-specific lncRNA *meiRNA* (see below) [21]. Thus, Ste11 functions as a major transcriptional regulator, acting precociously at the onset of the meiotic program.

Accurate control on the expression of Ste11 is fundamental to prevent the incidence of untimely meiosis. Beyond transcriptional repression by the GATA family Gaf1 factor [22], the induction of the *ste11*+ gene is also restricted during the mitotic cell cycle by the lncRNA *rse1* (repressor of *ste11* expression 1) [23]. *rse1* is a polyadenylated lncRNA of 2336 nucleotides in length that is transcribed divergently compared to the promoter of *ste11+* (Figure 1a). The deletion of *rse1* or inhibition of its transcription by CRISPR interference both lead to the accumulation of *ste11*+ mRNAs due to increased RNA PolII occupancy at the locus. *rse1* strictly acts in *cis*, since plasmid-borne versions of the transcript fail to repress *ste11*+ expression. Histone H3 lysine 14 acetylation (H3K14Ac) is also strongly enriched at the *ste11*+ promoter in cells lacking *rse1*. This epigenetic mark is specifically required for *ste11*+ derepression, as a H3K14R mutant defective for acetylation almost completely suppresses the effect of *rse1* deletion. These data suggested that *rse1* may cooperate with effector proteins to locally regulate the chromatin status of the *ste11*+ promoter and hence repress its expression in mitotic cells.

Biochemical approaches subsequently allowed the identification of the RNA-binding protein Mug187 as a prominent *rse1* partner [23]. ChIP experiments further demonstrated that Mug187 accumulates at the *ste11*+ promoter in a *rse1*-dependent manner. To gain additional mechanistic insights about the role of the *rse1*-Mug187 complex, a yeast two-hybrid screen was conducted and led to the identification of Lid2 (little imaginal discs 2) as a major Mug187 partner. Involved in the control of heterochromatic gene silencing, Lid2 is an essential H3K4me3 demethylase that contains a JmjC domain essential for its interaction with the H3K4 methyltransferase Set1 [24,25]. Highlighting the relevance of these interactions, *rse1* is required for the robust binding of both Lid2 and Set1 to the *ste11*+ promoter, similar to Mug187 [23]. Furthermore, the absence of these factors and of the Hos2 histone deacetylase (HDAC) that acts downstream of Set1 [26] leads to the derepression of *ste11*+. Together, these data support a model in which *rse1* recruits Mug187, Lid2 and Set1 to ensure the repression of *ste11*+ expression during vegetative growth through histone deacetylation at the promoter [23] (Figure 1a).

Further establishing *rse1* as a negative regulator of sexual differentiation, its deletion suppresses meiosis defects observed in the absence of Rst2, a transcriptional activator of *ste11*+ [23,27]. Interestingly, *ste11*+ induction upon nitrogen starvation is accompanied by a downregulation in *rse1* levels, possibly reflecting the molecular basis of a regulatory switch (Figure 1b). In line with this, it is noteworthy that the initial characterization of the *ste11+* locus revealed the existence of a second lncRNA, named *rce1* (*rse1*
control element 1), located immediately upstream *rse1* [23]. Whether *rce1* contributes to the regulation of *rse1* levels and hence *ste11*+ expression remains an open question, but it is possible that an intricate regulatory network operating locally between two distinct lncRNA species fine-tunes *ste11*+ expression in response to external cues. This is reminiscent of the situation in the evolutionarily distant budding yeast *Saccharomyces cerevisiae*, whereby expression of the key meiotic transcription factor *IME1* (Inducer of Meiosis 1) is epigenetically controlled by the upstream, albeit co-oriented, lncRNAs *IRT1* and *IRT2* (IME1 regulatory transcript 1 and 2, respectively) [28,29,30]. Proximal *IRT1* recruits chromatin-modifying enzymes to repress *IME1* transcription in haploid cells, while the accumulation of distal *IRT2* in sporulation-prone conditions interferes with *IRT1* induction, favoring in turn *IME1* expression and hence sexual differentiation [28,29,30].

## 3. The lncRNA *meiRNA* Lures the YTH-Family RNA-Binding Protein Mmi1 to Promote Expression of Gametogenic Transcripts upon Meiosis Onset

LncRNA-based post-transcriptional mechanisms also contribute to the fine tuning of meiotic gene expression, either promoting or inhibiting cell commitment to sexual differentiation. These involve dedicated RNA-binding proteins (RBPs) that shape transcriptomes, regulating RNA stability, export, translation and degradation, and hence mediate cell fate decisions. Among them is the YTH family RNA-binding protein Mmi1 (meiotic mRNA interceptor 1), which selectively targets a subset of meiosis-specific transcripts constitutively produced during mitosis for degradation by the nuclear exosome, thereby precluding untimely meiotic gene expression [31]. Mmi1 recognizes, via its C-terminal YTH domain, a *cis*-element within meiotic mRNAs named DSR (for determinant of selective removal) that is enriched in repeats of the hexanucleotide motif UNAAAC [31,32,33,34,35,36]. To limit the accumulation of DSR-containing meiotic transcripts in vegetative cells, Mmi1 cooperates with a number of cofactors, including the small nuclear protein Erh1, with which it assembles to form the heterotetrameric EMC (Erh1-Mmi1 complex) [37,38,39], components of the polyadenylation/termination machinery [40,41,42,43,44] as well as the multi-subunit MTREC (Mtl1-Red1 complex) that physically links RNA-bound Mmi1 to the nuclear exosome [45,46,47,48,49] (Figure 2a). Mmi1 colocalizes with these factors in scattered nuclear foci [31], suggestive of mRNA degradation factories, and further mediates nuclear retention of meiotic mRNAs to prevent their export to the cytoplasm [50]. Conversely, upon meiosis onset, Mmi1 activity in meiotic mRNA degradation must be efficiently inhibited. This is achieved by the lncRNA *meiRNA* that titrates Mmi1 away from its bona fide RNA targets during meiotic prophase I, thereby allowing the expression of meiotic program [31,51] (Figure 2b).

The *sme2+* gene (suppressor of *mei2*+), from which *meiRNA* is transcribed, was identified almost three decades ago as a high-copy suppressor of a thermosensitive mutant of *mei2*+, which encodes a key RNA-binding protein essential for pre-meiotic DNA replication and meiosis I [52]. Similar to Mei2, *meiRNA* is absolutely essential for sexual differentiation, as *sme2∆* cells fail to complete meiosis, most likely as a consequence of defective Mmi1 inhibition [31]. Detectable only in cells undergoing meiosis, *meiRNA* is produced as two polyadenylated isoforms, a shorter one of about 0.5 kb (named meiRNA_short_ or meiRNA-S) and an approximately 1 kb longer species (named meiRNA_long_ or meiRNA-L) [51,52,53]. Both Mei2 and Mmi1 bind to *meiRNA* in vivo and in vitro, the former preferentially associating with its 5′ region while the latter interacts with the 3′ part enriched in UNAAAC motifs [51,52]. Importantly, microscopy analyses showed that Mmi1-containing nuclear foci observed during vegetative growth redistribute to a single nuclear body during meiotic prophase I that co-localizes with Mei2 and overlaps the *meiRNA* transcription site (i.e., *sme2*+ gene) (Figure 2b) [31,51,53]. Formation of this structure, referred as the Mei2 dot, is critical for successful sexual differentiation. Indeed, deletion of the 3′ region of *meiRNA* or mutations of the UNAAAC motifs not only impair Mmi1 sequestration at the *sme2*+ gene but also prevent the accumulation of meiotic mRNAs and the completion of meiosis [51]. Based on these observations, it was proposed that *meiRNA* acts as a decoy to lure Mmi1, thereby allowing meiotic transcripts to escape degradation and thus favoring meiosis progression.

Akin to DSR-containing meiotic mRNAs, meiRNA is targeted for degradation by EMC, MTREC and the nuclear exosome during vegetative growth. It is noteworthy that one of the nuclear bodies in which these factors colocalize lies at the *sme2*+ locus, strongly suggesting that *meiRNA* decay occurs at or in the vicinity of its transcription site (Figure 2a). Upon meiosis onset instead, *meiRNA* accumulates and sequesters Mmi1 as part of the Mei2 dot (Figure 2b). How then does *meiRNA* escape Mmi1-dependent degradation in these conditions? Former work revealed that MTREC foci disassemble during meiotic prophase I, possibly alleviating a physical association between Mmi1 and the nuclear exosome and hence allowing *meiRNA* accumulation [54]. Consistent with this notion, the proline-rich protein Pir1 that physically connects Mmi1 to MTREC was recently shown to be targeted for proteolysis by the ubiquitin–proteasome system upon meiosis onset [55]. Highlighting the importance of this regulation, the expression of a Pir1 mutant escaping degradation prevents the full induction of *meiRNA* as well as the formation of the Mmi1–Mei2 dot [55]. Thus, the lack of a stable Mmi1–MTREC association is likely to underlie the accumulation of *meiRNA* during meiosis, which might in turn favor Mmi1 luring and expression of the meiotic program. In addition to regulated interactions between RNA decay effectors, the binding of Mei2 and/or additional meiosis-specific factors may also contribute to decrease *meiRNA* turnover, possibly stabilizing the transcript and/or limiting the accessibility of its 3′ end to exonucleolytic attack by the exosome. Future investigation will be required to unveil the precise mechanisms involved in the molecular switch underlying the modifications in *meiRNA* expression during the mitotic and meiotic cell cycles, and hence its function.

## 4. The lncRNA *mamRNA* Scaffolds the Antagonistic RNA-Binding Proteins Mmi1 and Mei2 to Exert Their Mutual Inhibition during Mitosis

Beyond its association with nuclear RNA degradation factors in mitotic cells, Mmi1 also tightly associates with the evolutionarily conserved Ccr4-Not complex [37,56,57,58,59], the major mRNA deadenylase involved in cytoplasmic mRNA turnover [60,61]. Surprisingly, we previously found that efficient meiotic mRNA degradation does not require the deadenylation activity of the complex but instead depends on the integrity of the E3 ubiquitin ligase subunit Not4/Mot2 [59]. We showed that in *mot2∆* mitotic cells, the accumulation of meiotic transcripts, including *meiRNA*, is mechanistically linked to the defective ubiquitinylation and downregulation of the meiosis inducer Mei2. These findings led to the proposal that the Mmi1-dependent meiotic RNA degradation pathway is under the control of a regulatory circuit that involves Mmi1 itself, which recruits Ccr4-Not to limit the accumulation of its own inhibitor Mei2 [59]. Maintaining low levels of Mei2 during mitosis is critical to safeguard Mmi1 activity, but how both proteins reciprocally inhibit their own activity in vegetative cells has remained elusive.

Recently, we have unraveled the mechanistic basis underlying the Mmi1–Mei2 mutual control. The Mmi1 and Mei2 RNA-binding activities, carried by their YTH and RRM3 domains, respectively, were first shown to be essential for their reciprocal regulation [62]. Single amino acid substitutions in the Mmi1 YTH domain indeed resulted in increased Mei2 levels, similar to cells lacking Mmi1 or Mot2. Conversely, the expression of a Mei2 mutant unable to associate with RNA (i.e., Mei2-F644A) failed to inactivate Mmi1 in *mot2∆* mitotic cells. These data suggested that the mutual regulation of Mmi1 and Mei2 may require an RNA species. To investigate this possibility, we determined the repertoire of transcripts simultaneously bound by both proteins in mitotic cells and identified several lncRNAs among the most enriched targets, including *meiRNA* (see above), *omt3* (see below) and a previously unannotated transcript produced upstream of the snoU14 gene, which we termed *mamRNA* for Mmi1 and Mei2-associated RNA [62]. Further characterization revealed that *mamRNA* is a bona fide lncRNA which accumulates as two abundant, mainly non-adenylated isoforms of roughly 550 and 700 nucleotides in length. Remarkably, despite the presence of three UNAAAC motifs, *mamRNA* is immune to Mmi1-dependent degradation during vegetative growth, contrary to other meiotic transcripts, including *meiRNA*. The lack of substantial polyadenylation and the robust accumulation of *mamRNA* in proliferating cells echoes the metazoan *MALAT1* and *NEAT1* lncRNAs, whose 3′ ends are produced by endoribonucleolytic cleavage and formation of a triple helix structure that acts as a degradation barrier [63]. Whether *mamRNA* is subjected to similar enzymatic and/or RNA folding events is an attractive hypothesis for future studies. As for *MALAT1* and *NEAT1*, which assemble with various proteins in subnuclear structures (e.g., nuclear speckles and paraspeckles) to regulate genome expression and integrity [63], they both accumulate in human germline cells and have been linked to reproductive disorders and cancer [64,65,66], suggestive of a role in gametogenesis.

Subsequent mechanistic analyses illuminated the role of *mamRNA* in the Mmi1–Mei2 mutual control during mitosis. The deletion of the lncRNA leads to increased Mei2 levels comparable to what is observed in *mot2∆* and *mmi1∆* cells [62]. This strongly supports a model in which Mmi1 associates with *mamRNA* to target Mei2 to Mot2 (Figure 2a), which resembles the role of the lncRNA HOTAIR in human cells that scaffolds RNA-binding domain-containing E3 ubiquitin ligases with their respective protein substrates to facilitate ubiquitinylation and proteasomal degradation [67,68]. Conversely, *mamRNA* is necessary for Mmi1 inactivation by high Mei2 levels, as its deletion suppresses the accumulation of meiotic transcripts in *mot2∆* cells. Together with single-molecule FISH experiments showing that *mamRNA* predominantly localizes to a nuclear dot overlapping one of the Mmi1-containing foci in both wild type and *mot2∆* cells (Figure 2a) [62], these results unraveled a major role for *mamRNA* in bringing Mmi1 and Mei2 in spatial proximity to fine tune meiotic gene expression in vegetative cells. However, and contrary to *meiRNA* during meiosis, *mamRNA* is unlikely to function as a decoy since meiotic transcripts are still efficiently bound by Mmi1 (yet accumulate) in the absence of Mot2 and persist in the nucleus [62]. Hence, Mmi1 inactivation by Mei2 and *mamRNA* in mitotic cells occurs downstream of RNA recognition and nuclear retention.

*mamRNA* thus defines a novel lncRNA involved in the reciprocal control of post-transcriptional effectors of gametogenesis, although detailed pictures about its biogenesis and mechanism of action are still lacking. Future work aimed at identifying regulatory *cis*-elements and *trans*-acting factors will undoubtedly provide insights about *mamRNA* biology and its peculiarity with respect to other Mmi1 targets.

## 5. Transcription Termination of the lncRNA *nam1* Protects Expression of the Downstream Gametogenic *byr2*+ Gene

Previous studies reported a role for RNA 3′ end processing and termination factors in the selective elimination of meiotic mRNAs during vegetative growth [40,41,42,43,54,69]. Several subunits of the cleavage and polyadenylation factor (CPF), the canonical mRNA poly(A) polymerase Pla1 and the conserved Rat1/XRN2 family 5′-3′ exoribonuclease Dhp1 indeed associate and co-localize with Mmi1, and mutants of these factors suppress sporulation defects due to the absence of *meiRNA*. CPF subunits and Dhp1 are also enriched at DSR elements, which are sites of premature transcription termination in Mmi1 targets coinciding with RNA Pol II pile-ups [44,69,70]. The overlap between Mmi1 and CPF/Dhp1 ChIP-seq profiles and the physical associations of CPF with both Mmi1 and the nuclear exosome together point to a role for Mmi1 in termination-coupled RNA degradation [44,70].

In addition to meiotic mRNAs, Mmi1 promotes the termination of various lncRNAs in mitotic cells, which is critical to prevent transcriptional interference of downstream genes. These include the *nc-tgp1* and *prt* lncRNAs that act as *cis* regulators of their neighboring phosphate metabolism-related *tgp1*+ and *pho1+* genes, respectively [33,71,72,73,74,75]. Another important Mmi1 target is the lncRNA *nam1* (for non-coding RNA associated to Mmi1) [34], transcribed upstream the *byr2*+ gene, which encodes a mitogen-activated protein kinase kinase kinase (MAPKKK) essential for meiosis entry (Figure 3a) [76,77]. *nam1* was initially discovered in RIP-seq experiments aimed at identifying novel Mmi1 RNA targets [34]. Akin to DSR-containing RNAs, *nam1* contains nine UNAAAC motifs predominantly localized at its 3′ end, and single base substitutions to generate GNAAAC sequences abolish its binding to Mmi1, resulting in its accumulation in vegetative cells [34]. Importantly, the expression of this mutated form of *nam1* (i.e., *nam1-1*) correlates with lower Byr2 protein levels and is associated with pronounced meiosis defects that are rescued by the ectopic expression of the *byr2*+ gene. Detailed analyses showed that this is due to *nam1* read-through transcription that invades the *byr2*+ locus (Figure 3b) [34]. The insertion of a potent transcription termination sequence immediately downstream of *nam1* indeed reduces the accumulation of read-through transcripts, restores *byr2*+ expression and almost completely suppresses sexual differentiation defects in *nam1-1* and *mmi1∆* cells [34]. Together with ChIP-seq experiments showing an Mmi1-dependent enrichment of CPF factors at the 3′ end of *nam1* [70], these data indicate that transcription termination of a lncRNA by Mmi1 prevents the repression of the downstream *byr2*+ gene and hence maintains the ability of cells to enter meiosis upon nutritional starvation (Figure 3a). Thus, Mmi1 also acts as a positive regulator of sexual differentiation.

Recent work further unveiled a mechanism by which lncRNAs, including *nam1*, mediate the repression of neighboring genes. Notably, read-through transcription allows lncRNAs to incorporate cryptic introns that are recognized by the conserved Pir2 protein and splicing factors, which in turn recruit gene silencing effectors, including the RNAi machinery and chromatin-modifying enzymes (Figure 3b) [78]. As such, mutations in *cis*-acting cryptic introns or of the above-mentioned *trans*-acting factors result in a substantial increase in the expression of downstream genes (e.g., *byr2*+). Importantly, sporulation defects due to *byr2*+ repression by increased *nam1* levels in cells lacking the exosome component Rrp6 [34] are rescued upon the mutation of Pir2 or splicing factors [78]. Thus, the regulation of *nam1* termination provides a highly flexible, lncRNA-dependent system for cells to control commitment to meiosis.

## 6. Homologous Chromosomes Pairing during Meiosis Requires the lncRNAs *meiRNA* and *omt3*

Following mating and conjugation, *S. pombe* cells transiently arrest in the G1 phase and undergo pre-meiotic DNA replication prior to karyogamy, whereby the nuclei of both parental cells fuse. The thus formed zygote enters meiotic prophase, during which the nucleus continuously oscillates back and forth between the cell poles, giving a specific elongated shape resembling a horse tail. Nuclear oscillation during the “horse-tail” stage promotes the pairing of homologous chromosomes, a pre-requisite for meiotic recombination and chromosome segregation [79,80,81,82]. This is facilitated by telomere clustering at the spindle pole body (SPB) and subsequent chromosome movement that increase contacts between homologous regions. Highlighting the biological importance of these motions, mutants deficient for telomere clustering or nuclear oscillation exhibit a dramatic decrease in recombination frequency, which in turn causes chromosome mis-segregation [83]. However, the mechanisms involved in the recognition of homologous chromosomes have remained elusive.

Insights came from the observation that individual Mei2 dots formed at the *meiRNA*-encoding *sme2*+ gene (see above) in mating haploid cells rapidly coalesce with one another upon nuclear fusion and persist within a unique dot during most of the horse-tail stage [84], reviewed in [85]. This indicated that the *sme2*+-containing genomic region displays robust pairing early upon meiosis onset, and consistent with this, pairing frequencies at other selected loci were significantly lower. Importantly, pairing was dependent on *meiRNA* transcripts originating from both homologous chromosomes and ectopic expression of the lncRNA was sufficient to confer a marked association of other genomic loci [84]. Dissection of the *cis*-elements involved in chromosome pairing further revealed that the 5′ region of *meiRNA* is dispensable, strongly suggesting that Mei2 binding is not required for this process (see above). On the contrary, the 3′ part of the transcript involved in Mmi1 sequestration [51] is necessary for the accumulation of *meiRNA* on the chromosome and efficient pairing of homologues [84]. Together, these data demonstrated that *meiRNA* acts as a *cis*-acting pairing factor, facilitating contacts between homologous chromosomes during meiotic prophase.

Microscopy-based screening subsequently identified several RNA-binding proteins colocalizing with the Mei2 dot, and hence *meiRNA*, during meiotic prophase [86]. Among these, six partake in chromosome pairing at the *sme2*+ locus, including the 3′ end processing/termination factors Seb1 and Rhn1 as well as the poly(A) binding protein Pab2. Interestingly, these Smp factors (for sme2 RNA-associated proteins) form at least two additional prominent nuclear foci that were mapped on chromosomes I and III, localizing to the meiosis-specific lncRNA transcription sites *omt3* and *lncRNA584*, respectively [86]. Similar to *meiRNA*, both lncRNAs accumulate at their genomic loci, but only *omt3* is critical for pairing of chromosomes I in a Smp-dependent manner, suggesting that additional features may contribute to the pairing of chromosomes III in the vicinity of *lncRNA584*. Live microscopy analyses of meiotic cells transiently exposed to 1,6-hexanediol, a compound that affects cellular structures exhibiting liquid–liquid phase separation properties [87,88], further gave insights about the biochemical nature of these chromosome-associated RNA assemblies. Individual lncRNA foci and loci pairing were rapidly disrupted upon 1,6-hexanediol treatment and gradually recovered following its removal, suggesting that Smp-lncRNA bodies may share common features with liquid droplets [86].

LncRNA-dependent chromosome pairing therefore emerges as an early mechanism during meiosis that confers specificity for homologous search and whose efficiency may overwhelm a DNA sequence scanning strategy. The local and transient accumulation of lncRNA-containing complexes, possibly through phase separation, provides a dynamic and reversible system for chromosome recognition. Such a system is likely to optimize the progression of subsequent meiotic events. It should be noted, however, that *omt3* is dispensable for sporulation, contrary to *meiRNA* [89]. The possibility that multiple lncRNAs, distributed all along chromosomes, contribute conjointly to pairing is an attractive hypothesis for future studies.

## 7. Widespread Antisense lncRNAs Underlie Global Genome Reprogramming during Meiosis

Former tiling array-based and strand-specific RNA-seq analyses unveiled extensive antisense transcription across the *S. pombe* genome [90,91,92,93]. Several loci harboring sense, poorly expressed meiotic genes exhibit high levels of antisense transcription during vegetative growth, suggesting the existence of regulatory mechanisms for pairs of sense and antisense RNAs that may account for the silencing of some meiotic genes in mitotic cells. Consistent with this notion, the ectopic expression of antisense transcripts impaired meiosis progression due to defective expression and function of cognate sense genes [92]. Interestingly, this type of regulation has been described in budding yeast: the *IME4* gene (inducer of meiosis 4), encoding an mRNA N6-adenosine methyltransferase important for sporulation, is repressed by the *cis*-acting antisense *RME2* lncRNA in haploid cells, thereby preventing the premature initiation of the meiotic program [94,95]. Conversely, *RME2* expression is suppressed in diploid cells, allowing *IME4* transcription and sexual differentiation [94,95]. Whether the fission yeast orthologue *ime4*+ contributes to meiosis progression and is regulated in a similar fashion remains, however, to be investigated.

Subsequent in-depth studies in *S. pombe* further refined the complexity and remodeling of transcriptomes upon sexual differentiation, uncovering thousands of antisense lncRNAs that are specifically induced during meiosis progression, especially at late stages [96,97]. Importantly, the increased expression of these antisense lncRNAs globally correlates with the downregulation of paired-sense mRNAs, thereby providing a molecular basis for the regulatory switch in cell type-specific programs. However, whether decreased levels of sense mRNAs are directly caused by increased antisense transcription or result from an effect of the lncRNA in *trans* remains to be addressed in most, if not all, cases. The production of antisense lncRNAs may also be inherently boosted when the expression of the sense mRNA is decreased, as a consequence of a leaky control on transcription. It should also be noted that the expression of many antisense lncRNAs does not necessarily anti-correlate with sense mRNA levels, indicating the existence of intricate rules and mechanisms governing the relationship between lncRNA/mRNA pairs.

Meiotic antisense lncRNAs are preferentially, but not solely, targeted for degradation by the conserved cytoplasmic 5′–3′ exoribonuclease Xrn1/Exo2 and/or the dicer-family nuclear RNAi component Dcr1 [96,97]. Whether pairs of sense/antisense transcripts may, at least transiently, form RNA duplexes in vivo is still a subject of debate. While the formation of dsRNAs could provide a mechanism for the regulation of mRNA stability, export and translation, it is also likely that many transcripts from opposite strands do not coexist in the same cell. Regardless of these considerations, antisense transcription is a universal and powerful system for gene regulation [98]. Due to their prevalence, meiotic antisense lncRNAs may constitute a considerable reservoir of regulatory molecules, each having the potential to modulate gene expression in different ways and at specific stages along meiosis progression. The study of individual cases will certainly fill these conceptual gaps by illuminating their mechanism of action and biological relevance to the intricate control of sexual differentiation.

## 8. Conclusions and Perspectives

Fission yeast gametogenesis is a highly regulated process involving a combination of chromatin-based, transcriptional, post-transcriptional and post-translational mechanisms that repress cell commitment to meiosis during vegetative growth or conversely, promote sexual differentiation upon nutritional starvation. LncRNAs take their part in this developmental switch by modulating the expression of meiotic genes and/or fine-tuning the activities of gametogenic effectors. Since its initial discovery and for more than two decades, *meiRNA* has been the sole example of a regulatory lncRNA involved in the control of *S. pombe* meiosis. Only recently, thanks to the development of high-throughput sequencing technologies, the list of non-coding transcripts with an assigned function has expanded, including *rse1*, *mamRNA*, *nam1* and *omt3*. Given the prevalence of lncRNAs in the genome and among transcripts induced during meiosis progression, additional players will undoubtedly enrich this catalogue in the near future. It is remarkable that the above-mentioned lncRNAs cover a broad range of modes of action, from guiding chromatin-modifying enzymes to specific genomic loci, sequestering or scaffolding RNA-binding proteins, regulating the expression of neighboring genes, to promoting homologous chromosome pairing. However, this is probably only the tip of the iceberg and it is tempting to speculate that the regulation of additional processes such as pre-meiotic DNA synthesis, telomere clustering and meiotic divisions involves lncRNAs as well. This also highlights the power and utility of fission yeast as a model organism to explore the diversity of lncRNA functions and to obtain valuable knowledge on the mechanisms involved in the control of germ cell differentiation.

Even though great progress has been made over the last few years, lncRNA biology in general, and more specifically in the context of gametogenesis is only at its beginning. What are the expression profiles of regulatory lncRNAs during the mitosis to meiosis transition? How do modulations in their levels, if any, translate into functional outcomes? Are they structured, and if so, what are the *cis*-elements involved and how is this influenced by protein partners? How do they contribute to the assembly of large protein-RNA bodies? Are they targeted for post-transcriptional modifications? In this case, what are the enzymes involved and the proteins that read these marks? Which lncRNAs engage productive translation to generate functional peptides? Future investigation in fission yeast should provide answers to these fascinating questions, allowing a better understanding of the role of lncRNAs and possibly uncovering evolutionarily conserved principles in the regulation of germ cell differentiation.

## Figures and Tables

**Figure 1 ncrna-07-00034-f001:**
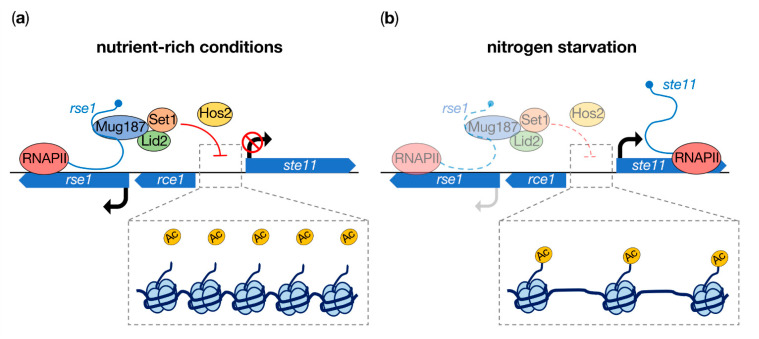
*rse1*-dependent repression of *ste11*+ involves histone deacetylation. (**a**) In nutrient–rich conditions, the lncRNA *rse1* recruits Mug187, Lid2 and Set1 to promote histone 3 lysine 14 (H3K14) deacetylation by Hos2 at the promoter of *ste11*+, thereby preventing its induction. An array of hypoacetylated nucleosomes upstream the *ste11*+ gene is depicted in the insert. (**b**) Upon nitrogen starvation, decreased levels of *rse1* favor transcription of *ste11*+, most likely as the consequence of lower nucleosome occupancy and higher H3K14 acetylation in the gene promoter. Whether the lncRNA *rce1* inhibits expression of *rse1* in this context remains to be investigated.

**Figure 2 ncrna-07-00034-f002:**
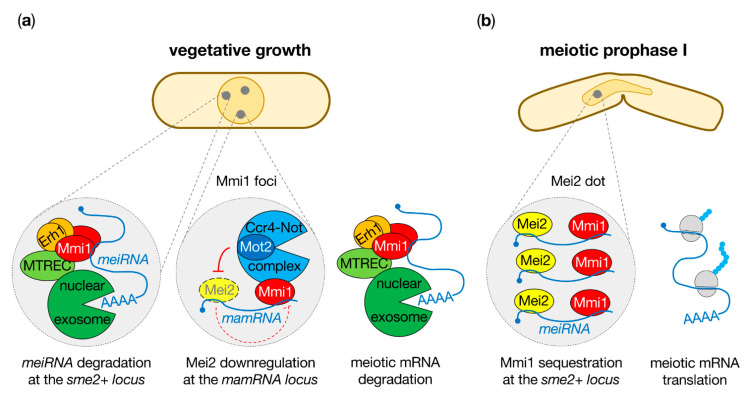
The lncRNAs *meiRNA* and *mamRNA* modulate the activities of the RNA-binding proteins Mmi1 and Mei2 to tune meiotic gene expression. (**a**) During vegetative growth, Mmi1 targets DSR-containing meiotic RNAs, including *meiRNA*, for degradation by the nuclear exosome. Mmi1 also associates with *mamRNA* to target Mei2 for ubiquitinylation and downregulation by the E3 ubiquitin ligase subunit Mot2 of the Ccr4-Not complex (see Section 4). Two of the scattered Mmi1 nuclear foci overlap with *meiRNA* and *mamRNA* transcription sites. The localization of the additional foci is not known. Whether meiotic mRNA degradation occurs within these structures also remains to be investigated. (**b**) During meiotic prophase I, Mmi1 is sequestered by Mei2 and *meiRNA* in the so-called Mei2 dot that lies at the *meiRNA* transcription site (i.e., *sme2*+ gene). This allows meiotic mRNAs to escape degradation and enter translation, thereby ensuring meiosis progression.

**Figure 3 ncrna-07-00034-f003:**
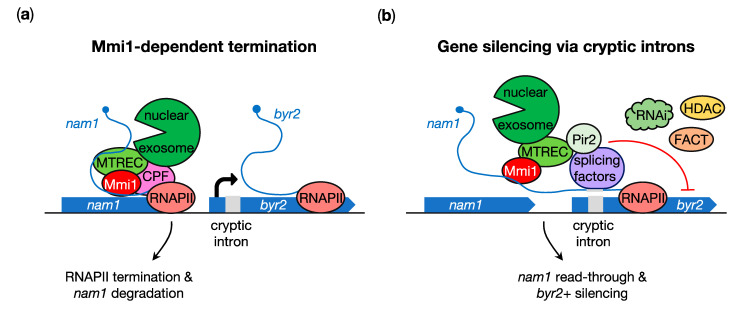
Control of *byr2*+ expression by the lncRNA *nam1*. (**a**) Mmi1 associates with components of the cleavage and polyadenylation factor (CPF) to mediate RNAPII termination upstream the *byr2*+ gene and targets the DSR-containing lncRNA *nam1* for degradation by the nuclear exosome. (**b**) Upon environmental changes, *nam1* transcription extends into *byr2*+ and the lncRNA incorporates a cryptic intron that is recognized by splicing factors and Pir2, which in turn recruit silencing effectors (e.g., the RNAi machinery, the chromatin-remodeling FACT complex and the histone deacetylase Clr3) to repress *byr2*+ expression.

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
