# Peer review of "Long Non-Coding RNAs in the Control of Gametogenesis: Lessons from Fission Yeast"

_ncrna, 2021, doi:10.3390/ncrna7020034_

Round 1

Reviewer 1 Report

The review from Andric and Rougemaille builts on the recent contribution of their lab in this field. To my knowledge, this is the first attempt to review the fission yeast literature on this hot topic. The review is very well-written and presents all the numerous actors in an easy to follow manner. The well-characterized examples of ncRNAs in the control of gametogenesis are presented in a very balanced manner with adequate emphasis on the work of the author. The review presents an informative and synthetic view of the outstanding contribution of fission yeast to our understanding of the role of ncRNA in the control of gametogenesis. It also reminds the pioneer work of Yamamoto on meiRNA, which is not always recognized as it should. To me this important review can be published as such.

Reviewer 2 Report

In the current manuscript, Andric and Rougemaille have reviewed the role of lncRNAs in the regulation of fission yeast meiosis. The authors summarize current knowledge of lncRNAs rse1, meiRNA and mamRNA function in detail. A focus is on the mechanistic basis of mamRNA dependent regulation of the RNA binding proteins Mmi1 and Mei2, recently discovered in the authors’ lab (Nat Commun. 2021, doi: 10.1038/s41467-021-21032-7). The paper is well written and highlights important novel insights into non-coding RNA function.

Specific comments for improvement/clarification:

The paper specifically addresses the role of lncRNA in regulation of fission yeast gametogenesis. To increase the relevance for a broader readership, some brief comments about other model systems might be useful. For example, it is not mentioned whether gametogenesis in S. cerevisiae is similarly regulated by lncRNA or whether the lncRNAs and/or their target proteins are evolutionary conserved. Also, some more information about metazoan lncRNA might be useful (see below).

Line 74: reword, e.g.: „While certain functions are essential to ...”

Figure 1 and Figure 2: There seems to be a font problem that should be fixed.

Line 246: metazoan MALAT1 and NEAT1 lncRNAs are briefly mentioned due to their similarity to mamRNA, but their cellular functions are not explained. Do they resemble mamRNA only with respect to lack of polyadenylation or are they related to meiosis regulation as well?

Line 281: “sites of premature termination”. It might be useful to clarify that this refers to transcriptional termination – could be confused with an effect on the translation of Mmi1 target mRNAs
